# Understanding Myelodysplasia and Inflammation Through the Lense of VEXAS Syndrome: A Review

**DOI:** 10.3390/cells13221890

**Published:** 2024-11-15

**Authors:** Louis Wolff, Leo Caratsch, Lin-Pierre Zhao, Sabine Blum, Denis Comte

**Affiliations:** 1Department of Internal Medicine, Erasme Hospital, Brussels University Hospital, Université Libre de Bruxelles (ULB), Route de Lennik 808, 1070 Brussels, Belgium; louis.wolff@hubruxelles.be; 2Department of Medicine, Division of Internal Medicine, Lausanne University Hospital (CHUV), University of Lausanne, 1005 Lausanne, Switzerland; 3Assistance Publique des Hôpitaux de Paris (AP-HP), Department of Hematology, Hôpital Saint-Louis, Université de Paris, 75010 Paris, France; 4Division of Hematology and Hematology Central Laboratory, Lausanne University Hospital (CHUV), University of Lausanne, 1005 Lausanne, Switzerland

**Keywords:** MDS, UBA1, VEXAS, inflammation, myelodysplastic syndromes, dysplasia, auto-inflammatory, auto-immune

## Abstract

VEXAS syndrome, a monogenic X-linked disorder resulting from mutations in the UBA1 gene, has emerged as a key model for unraveling the links between systemic inflammatory or autoimmune diseases (SIAD) and myelodysplastic syndromes (MD). This syndrome is characterized by the presence of vacuoles, X-linked inheritance, autoinflammation, and somatic mutation patterns, highlighting a unique intersection between genetic and immunological dysregulation. Apart from VEXAS, 10% to 30% of individuals diagnosed with MDS exhibit SIAD phenotypes, a significant increase compared to the 5% incidence in the general population. In this comprehensive review, we aim to elucidate the molecular mechanisms driving the pro-inflammatory environment in MDS, focusing on the contribution of VEXAS syndrome to this complex interplay. We examine how UBA1 mutations disrupt cellular homeostasis, triggering inflammatory pathways. Furthermore, we explore the broader implications of these findings for the pathogenesis of MDS, proposing that the inflammatory dysregulation of VEXAS may shed light on mechanisms of disease progression and identify potential therapeutic targets in MDS. Through an integrated analysis of genetic, immunological, and clinical data, this review seeks to deepen our understanding of the complex relationship between systemic inflammation and hematological malignancies, paving the way for new diagnostic and therapeutic strategies.

## 1. Introduction

VEXAS syndrome (Vacuoles, E1 enzyme, X-linked, Auto-inflammatory, and Somatic), first described in 2020, is a monogenic disease resulting from a mutation in the UBA1 gene. This condition is characterized by a unique combination of systemic inflammatory and auto-immune disease (SIAD), along with hematological disorders, predominantly myelodysplastic syndromes (MDS) [1]. Apart from VEXAS syndrome, around 10 to 30% of patients with an MDS present with SIAD features, compared to only 5% in the general population [2]. In this review article, we will explore the mechanisms underlying the pro-inflammatory state in myelodysplastic syndromes with a particular focus on VEXAS syndrome.

## 2. Pro-Inflammatory State in Myelodysplastic Syndrome

MDS are blood disorders arising from clonal changes in hematopoietic stem cells (HSC). These alterations lead to ineffective hematopoiesis, characterized by dysplastic changes that result in cytopenias. Most individuals with MDS exhibit gene mutations affecting various gene classes, including epigenetic regulators (e.g., TET2, ASXL1), spliceosome genes (e.g., SF3B1, SRSF2), transcription factors (e.g., TP53, RUNX1), genes associated with signaling pathways (e.g., KRAS, JAK2), and components of the cohesin complex [3]. The accumulation of these mutations provides the host cells with a proliferation advantage, resulting in cytopenias and associated disease manifestation, including hemorrhages, infections, and asthenia, as well as an increased risk of transformation into acute myeloid leukemia (AML) [4].

MDS are often associated with SIAD. Epidemiological data reveal that between 10% and 30% of individuals diagnosed with MDS have SIAD phenotypes, suggesting a possible link between inflammation and the pathogenesis of certain hematological disorders [5]. The genetic immune landscape of SIAD and MDS reveals polymorphisms that link these conditions. Notably, HLA-B27, an MHC class I molecule that is essential for presenting peptides to T lymphocytes, is strongly associated with both spondyloarthropathies and hematological malignancies [6]. Additionally, polymorphisms in genes involved in cytokines signaling pathways can influence immune function, potentially leading to autoimmunity and hematological malignancies. These variations can affect the production, release, or response to cytokines, thereby disrupting the balance of pro- and anti-inflammatory signals. For example, variations in the gene encoding the IL1-receptor antagonist have been identified as risk factors for SIAD such as systemic lupus erythematous or myasthenia gravis, as well as for hematological malignancies [7,8,9,10] (Table 1).

In MDS, the cytokine profile indicates an upregulation of pro-inflammatory pathways, including a significant reduction in regulatory T cells (Tregs), which are essential for maintaining self-tolerance. Kordasti et al. observed this reduction in Tregs, along with an increase in Th17 cells, in patients with low-risk MDS [11,12]. This dysregulation is further exacerbated by an increase in the production of pro-inflammatory cytokines, such as interleukin-6 (IL-6) and tumor necrosis factor-alpha (TNF-α), which may have a pro-apoptotic effect on hematopoietic stem cells, contributing to the quantitative and functional impairment of Tregs and the expansion of Th17 cells [12,13,14,15,16]. Moreover, Ten-eleven translocation-2 (TET2) is crucial in maintaining immune balance by actively suppressing the transcription of IL-6 during the resolution of inflammation in innate myeloid cells, such as dendritic cells and macrophages. When this regulatory function is compromised, it can lead to the persistence of an activated pro-inflammatory state, a key feature in the pathophysiology of certain hematological disorders, including MDS-associated SIAD [17,18,19].

To further understand the upregulation of pro-inflammatory pathways in MDS, a French team examined the T cell phenotype in MDS patients with and without SIAD, observing a decreased Treg count and reduced cell surface expression of immune checkpoint inhibitors (ICR), both of which are crucial for maintaining self-tolerance in patients with SIAD. Genotypic analysis revealed that these SIAD patients had higher expression of TET2, Isocitrate dehydrogenase 1 gene (IDH), Serine/arginine-rich splicing factor 2 (SRSF2), and co-occurring TET2/IDH mutations. To explore the relationship between the genotype and phenotype, they compared SIAD patients with and without these mutations. Patients with a mutation in the gene were associated with a decrease in the central memory Treg (TCM) subset. Patients with TET2/IDH mutations exhibited a lower number of Tregs, central memory Treg subset, CD8+ stem cell memory, and transitional memory cells. Moreover, the TET2/IDH mutation was linked to decreased expression of CD96, a co-inhibitory cell surface receptor that is highly expressed by central memory T cells. Additionally, TET2 is important for the stabilization of Foxp3 expression, the canonical transcription factor for Treg function. Mutations of TET2 can lead to the downregulation of Foxp3, resulting in impaired Treg function and the development of autoimmunity. Interestingly, the deficit in TET2 activity and the resulting downregulation of Foxp3 can be restored with vitamin C supplementation, offering a potential therapeutic approach [20,21]. Decreased Treg function in MDS patients with SIAD may also be linked to elevated levels of IRF-1 (interferon regulatory factor-1), a transcriptional regulator that inhibits Foxp3 expression. Higher IRF-1 levels have been observed in MDS patients with SIAD, further contributing to the dysfunction of Tregs in this context [18].

γδ T cells are non-conventional T lymphocytes that play a critical role in recognizing and eliminating stressed or transformed cells, including auto-reactive cells. When γδ T cells become dysfunctional or impaired—whether due to a decrease in their numbers or reduced functional capacity—they may fail to control these auto-reactive cells effectively. This breakdown in immune surveillance can significantly contribute to the loss of self-tolerance observed in patients with MDS-associated SIAD, potentially exacerbating the inflammatory and autoimmune manifestations of the disease (Figure 1) [22,23]. Some studies also associate inflammation among MDS patients with NLRP3 inflammasome activation. This suggests potential targets for anti-IL-1 therapies [24].

## 3. Pro-Myelodysplastic State in Inflammation

Conversely, certain aspects of the inflammation process could contribute to the development of MDS. Some SIAD appear many years before the onset of an MDS, suggesting that they may represent a risk factor for the disease. A study based on the US Surveillance Epidemiology and End Results (SEER) database found a higher risk of developing acute myeloid leukemia (AML) (OR 1.29) and an MDS (OR 1.50) among patients with auto-immune diseases. Specifically, AML was associated with rheumatoid arthritis (OR 1.28), systemic lupus erythematosus (OR 1.92), polymyalgia rheumatica (OR 1.73), autoimmune hemolytic anemia (OR 3.74), systemic vasculitis (OR 6.23), ulcerative colitis (OR 1.72), and pernicious anemia (OR 1.57). Myelodysplastic syndrome was associated with rheumatoid arthritis (OR1.52) and pernicious anemia (OR 2.38) [25]. Inflammation, even caused by infection, may be a risk factor for developing MDS (Table 1) [26].

## 4. Comparison Between MDS with and Without SIAD

In terms of peripheral blood characteristics, patients with SIAD tend to exhibit a higher absolute neutrophil count in univariate analysis (OR 1.06, 95% CI [1.01; 1.12]) [27]. Interestingly, autoantibodies were detected in 50% of MDS patients without SIAD, possibly due to the inherent tendency of MDS to present with auto-inflammatory features, which are primarily driven by dysfunction in innate immunity rather than adaptative immunity [28,29]. Recent studies have highlighted significant molecular and clinical differences between MDS patients with and without SIAD. Lower-risk MDS, a subset of MDS characterized by a better prognosis, slower progression, fewer blast cells in the bone marrow, and milder cytopenias, are more frequently associated with SIAD patients. These lower-risk cases have a reduced likelihood of transforming into acute myeloid leukemia (AML) compared to higher-risk MDS. In multivariate analysis, SIAD MDS also presented with fewer blasts (OR 0.86, 95% CI [0.78; 0.93], *p* < 0.01). Even though karyotypes seem similar in MDS with or without SIAD, MDS associated with pseudo-Behçet syndrome have been described with trisomy 8 [30]. Furthermore, TET2 mutations are more prevalent among SIAD MDS patients (46% vs. 34%, *p* = 0.04), although the variant allele frequency (VAF) does not differ significantly between the groups [27]. This finding is supported by another study that showed that TET2-mutated MDS patients have a higher incidence of autoimmune features (31.3% vs. 5.3%, *p* = 0.001) [31]. Additionally, IDH1/2 mutations are more frequent in the SIAD MDS cohort (14% vs. 4%, *p* < 0.01), with a higher VAF being observed in SIAD patients (38% vs. 26%, *p* = 0.04). SRSF2 mutations are more common in SIAD MSD patients (31% vs. 15%, *p* < 0.01). These molecular markers, particularly TET2/IDH (OR 1.87, 95% CI [1.08; 3.2], *p* = 0.02) and SRSF2 (OR 2.21, 95% CI [1.14; 4.28], *p* = 0.02), underscore the unique genetic landscape of SIAD MDS and may contribute to their distinct clinical presentation and prognosis [27].

Data on overall survival (OS) in MDS patients with and without SIAD are inconclusive. While some studies suggest that OS is lower in SIAD MSD patients, others report no significant differences. A review of multiple studies presents mixed results, likely due to the retrospective nature of most studies, which introduces potential bias (Table 2). Furthermore, the heterogeneity of SIAD manifestations further complicates the interpretation of outcomes. Over the last decade, significant advancements in the treatment of both SIAD and MDS have added another layer of complexity to comparing survival outcomes across different patient cohorts.

## 5. VEXAS Syndrome

VEXAS syndrome belongs to a group of diseases known as proteasomopathies, which are characterized by abnormalities in the function or regulation of the proteasome, a cellular structure responsible for degrading proteins that have been tagged with ubiquitin. Ubiquitylation, a reversible post-translational modification, is crucial for regulating cellular responses to immune signals [36]. VEXAS syndrome is caused by mutations in the UBA1 gene, leading to dysfunction of the ubiquitin-like-modifier-activating E1 enzyme, the most important enzyme in the ubiquitylation cascade [1]. UBA1’s function cannot be compensated by another enzyme, and its impairment results in the accumulation of proteins. This protein aggregation triggers the unfolded protein response (UPR) and activates several inflammatory pathways, such as NF-κB and interferon pathways, resulting in a pro-inflammatory state [36]. Clinically, VEXAS syndrome primarily affects men and is marked by systemic inflammatory and autoimmune diseases (SIAD), with symptoms including systemic inflammation, hematologic abnormalities (including macrocytic anemia, thrombocytopenia, and vacuoles in myeloid and erythroid precursor cells), recurrent skin rashes, pulmonary manifestations, chondritis, and arthritis (Table 3) [36].

### 5.1. Is the MDS Homeostasis in VEXAS Similar to Other MDS?

A French cohort study on VEXAS syndrome by Georgin-Lavialle et al. found that 24% of the 75 MDS had additional somatic mutations, predominantly in DNMT3A and TET2. Similarly, a study by the NIH/Mayo Clinic reported clonal hematopoiesis in 80 VEXAS patients with or without MDS. Among them, 35% had one non-UBA1 clonal hematopoiesis mutation, and 25% had two or more. These findings confirmed the results from the French cohort, with most mutations affecting the DNMT3A and TET2 genes [42]. In the bone marrow, VEXAS patients typically present with vacuoles (also sometimes observed in peripheral blood), hypercellularity, granulocytic hyperplasia, mild dyspoiesis, usually a normal karyotype, and fewer blasts. MDS associated with VEXAS syndrome often have a lower risk profile and tend to involve fewer mutations than MDS without VEXAS [37]. In a cohort of patients with MDS, 1% presented with UBA1 mutations, and 7% lacked established myeloid mutations or disease classifications. This suggests the inclusion of UBA1 mutation research in all MDS cases [43].

The precise relationship between the UBA1 clone and the development of MDS remains uncertain. It is unclear whether the UBA1 clone directly initiates the events that lead to MDS or if the myeloid neoplasm develops because of the inflammatory microenvironment driving clonal selection. It is worth noting that various diseases characterized by high levels of inflammation, such as giant cell arteritis, periarteritis nodosa, Behçet, and others, may exhibit a lower tendency to induce MDS than VEXAS [44].

The overall risk profile of MDS, estimated using the IPSS (International Prognostic Scoring System) or R-IPSS (Revised International Prognostic Scoring System), was examined by Ferrada et al., who found that 96% of 23 patients had a low- or very-low-risk profile to develop AML [43]. A Swiss cohort of 17 patients, two-thirds of whom had had MDS, reported no cases of increased blasts in the bone marrow [41]. To date, only a few cases of MDS associated with VEXAS have progressed to AML. VEXAS patients, according to IPSS scores, generally have a lower risk profile for MDS compared with those not associated with VEXAS. Since VEXAS primarily affects older patients with a median life expectancy of 10 years, the duration of the inflammatory state may not be sufficient to promote the development of AML. Despite being classified as having a low risk of MDS, VEXAS patients often experience severe cytopenia and a significant need for transfusions. This has led some authors to characterize VEXAS as “a highly inflammatory clonal cytopenia” [37,39].

### 5.2. Therapeutic Approach

Concerning higher-risk MDS, a subset of MDS characterized by a worse prognosis, faster progression, more blast cells in the bone marrow, and severe cytopenias, 5-azacytidine (AZA) is one of the main treatments. This nucleoside analog chemotherapy works by inhibiting DNA methylation. It is administered via injection and is often prescribed to patients who are not candidates for stem cell transplantation. Patients treated with AZA may experience myelosuppression, leading to cytopenias, which can sometimes limit its use. Moreover, AZA may induce inflammatory SIAD, such as neutrophilic dermatosis. A study involving patients with SIAD and MDS found that 19 out of 22 individuals treated with AZA experienced positive effects on SIAD symptoms, with most being able to discontinue their immunosuppressive therapy as a result [45]. Research on AZA’s impact demonstrated that, although treatment increased Treg levels, these cells lacked the ability to proliferate. In vitro studies further revealed that 5-azacytidine reduced the suppressive function of CD4+ regulatory T cells and increased the production of interleukin-17 [46].

For VEXAS, the current consensus treatment involves corticosteroids combined with an additional immunosuppressive agent, such as anti-IL6, or JAKi (Janus kinase inhibitors). JAKi, such as ruxolitinib, have shown both anti-inflammatory and potential anti-clonal effects in studies involving myelofibrosis patients, although conclusive evidence of their anti-clonal benefits in VEXAS is still lacking. Surveillance of VAF levels during ruxolitinib treatment for UBA1 mutations has failed to show any notable decrease [47]. In a retrospective study on 110 patients evaluating the biotherapeutic efficacy, patients were defined as responders when either a complete response (clinical remission, C-reactive protein ≤10 mg/L, and ≤10 mg/day of prednisone-equivalent therapy) or a partial response (clinical remission and a 50% reduction in CRP levels and glucocorticoid dose) was completed. JAKi and anti-IL6 resulted in responders with rates of 30% and 26%, respectively, at 6 months [48]. Clinical studies have suggested that AZA may be capable of inducing complete molecular clearance of UBA1 mutations [49]. IDH1 and IDH2 inhibitors, such as Ivosidenib and Enasidenib, have shown promising results in treating MDS with mutations in these genes. Since these mutations are also implicated in both MDS and VEXAS syndrome, IDH inhibitors may offer potential therapeutic benefits for these conditions [50].

### 5.3. Is There a Place to Treat the Clone in VEXAS Without MDS?

Comont et al. reported on five patients with UBA1 mutation but without diagnostic criteria for a myelodysplastic syndrome treated with AZA with a very good outcome. All five patients achieved a complete inflammatory response, with no relapses being observed over a follow-up period ranging from 4 to 37 months, indicating a very favorable outcome [51].

## 6. Conclusions

VEXAS syndrome represents a unique intersection of genetic and inflammatory pathophysiology, primarily driven by UBA1 gene mutations. This syndrome illustrates the complex interplay between systemic auto-inflammatory and autoimmune diseases (SIAD) and hematological disorders, particularly myelodysplastic syndromes (MDS). VEXAS patients exhibit distinct clinical features, including a higher prevalence of specific somatic mutations (DNMT3A, TET2), unique bone marrow characteristics, and a generally lower risk of MDS progression. However, despite this lower risk, patients suffer from significant cytopenias and have high transfusion needs due to the syndrome’s inflammatory nature.

Treatment strategies for VEXAS differ from typical MDS therapies, frequently involving corticosteroids and immunosuppressive agents. AZA offers significant promise, particularly with its ability to induce complete molecular clearance of UBA1 mutations, making it a promising therapeutic option for managing both the hematological and inflammatory aspects of VEXAS syndrome. For selected young and fit patients, allogeneic stem cell transplantation should be considered, as it remains the only curative option. The persistent inflammatory state in VEXAS not only worsens hematologic dysfunction but also underscores the bidirectional relationship between inflammation and clonal hematopoiesis. Further research is critical to fully understand the mechanisms underlyi25ng this relationship and to develop targeted therapies that effectively address both the inflammatory and hematologic components of VEXAS syndrome.

## Figures and Tables

**Figure 1 cells-13-01890-f001:**
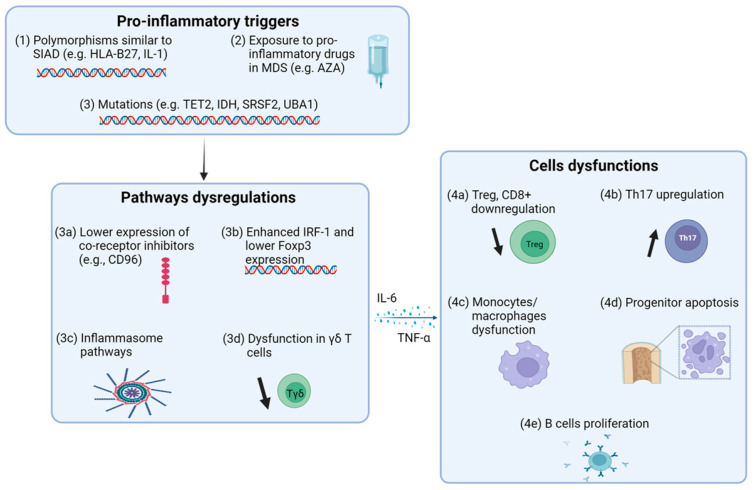
Overview of pro-inflammatory homeostasis in MDS. (1) Genetic polymorphisms, such as HLA-B27 and IL-1, are shared between SIAD and MDS patients. (2) Some drugs used to treat MDS have pro-inflammatory effects. (3) Mutations identified in MDS patients predispose them to a pro-inflammatory state, leading to (3a) reduced expression of co-inhibitory receptors, such as CD96, (3b) increased IRF-1 and decreased Foxp3 expression, (3c) upregulation of inflammasome pathways, and (3d) dysfunction of γδ T cells, a subset of non-conventional T lymphocytes that are crucial for recognizing and eliminating stressed or transformed cells, including auto-reactive cells. This pro-inflammatory homeostasis ultimately results in (4a) downregulation of Tregs and CD8+ T cells, (4b) upregulation of Th17 CD4+ T cells, which are key players in inflammation and development of autoimmunity, (4c) dysfunction of monocytes and macrophages, (4d) pro-apoptotic effects on hematopoietic stem cells mediated by pro-inflammatory cytokines, such as interleukin-6 (IL-6) and tumor necrosis factor-alpha (TNF-α), and (4e) increased B cell proliferation (MDS: myelodysplastic syndrome).

**Table 1 cells-13-01890-t001:** List of pro-inflammatory factors in the MDS environment and pro-MDS factors in AI diseases (MDS: myelodysplastic syndrome, SIAD: systemic inflammatory or autoimmune diseases).

	Factors Contributing to the Pro-Inflammatory State in MDS	Factors Contributing to MDS in the Pro-Inflammatory State
Temporality	MDSs may precede the diagnosis of SIAD	SIAD may precede the diagnosis of MDS
Genetic environment	Both SIAD and MDS share a similar genetic background (e.g., HLA-B27 or IL-1 polymorphisms)	
Homeostasis	The MDS environment is pro-inflammatory, with elevated levels of pro-inflammatory cytokines (e.g., IL-1, Th17) and mutations predisposing one to tolerance dysfunction (e.g., TET2/IDH mutations with less Treg and immune checkpoint inhibitors)	Inflammation, even caused by infection, may increase the risk for MDS
Treatment induced	Drugs used to treat MDS may trigger SIAD manifestations (e.g., Azacytidine)	Drugs used to treat inflammation may predispose one to MDS.
Improvement under treatments	In MDS with SIAD features, treatment of MDS may alleviate SIAD manifestations	
Impaired Immune Function	MDS are associated with immune dysfunction, resulting in a higher susceptibility to infections. Chronic infections can contribute to inflammation	

**Table 2 cells-13-01890-t002:** Comparison of overall survival (OS) among MDS patients with and without SIAD. OS showed no significant differences or was improved in patients with SIAD, except in the study by de Hollanda et al., where OS was worse in patients with vasculitis and cryoglobulinemia (Giannouli et al. [32], Hollanda et al. [33], Mekinian et al. [34], Komrokji et al. [35]).

Authors	Number of Patients (MDS vs. MDS/AI)	Country	Study	Outcome
Giannouli et al. [32]	57 vs. 13	Greece	Prospective	No difference
Hollanda et al. [33]	189 vs. 46	France	Retrospective	No difference overall but vasculitis subgroup with reduced OS
Mekinian et al. [34]	660 vs. 123	France	Retrospective	No difference
Komrokji et al. [35]	1408 vs. 391	UK, USA	Retrospective	OS were 60 months (95% CI, 50–70) for patients with SIAD vs. 45 months (95% CI, 40–49) without (*p* = 0.006)

**Table 3 cells-13-01890-t003:** Comparison of patient manifestations in VEXAS syndrome vs. MDS/CMML/SIAD (* Ferrada et al. [37], ^%^ Maeda et al. [38], ^£^ Georgin-Lavialle et al. [39], ^^^ Mascaro et al. [40], ^°^ Wolff et al. [41], ^#^ Mekinian et al. [34], MDS: myelodysplastic syndrome, SIAD: systemic auto-inflammatory and auto-immune disease, CMML: chronic myelomonocytic leukemia).

Manifestations	VEXAS *^%£^°^	MDS/CMML with SIAD ^#^
Median age, years, IQR	64–74	70
Male sex, %	96–100	67
Fever, %	64–92	35
Skin, %	72–84	55
Pulmonary, %	46–67	17
Chondritis, %	10–54	-
Adenopathy, %	35–47	-
Joint, %	47–73	70
Ocular, %	23–57	16
Peripheral nervous system, %	6–15	12
Heart, %	11–12	-
Kidney involvement, %	6–10	10
Venous thromboembolism, %	35–60	-
Myelodysplastic syndrome, %	31–71	100
Macrocytic anemia, %	71–97	-
Vacuoles, %	73–100	-

## Data Availability

Not applicable.

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
