# Peer review of "Understanding Myelodysplasia and Inflammation Through the Lense of VEXAS Syndrome: A Review"

_cells, 2024, doi:10.3390/cells13221890_

Round 1
Reviewer 1 Report
Comments and Suggestions for Authors
In this well written review by Wolff et al. the authors analyze the relationship between systemic inflammatory or autoimmune diseases and myelodysplastic syndromes (MDS) using VEXAS syndrome as a framework. This review is relevant and timely since the interest in the role of inflammation in clonal hematopoiesis, MDS and the development of hematological malignancies has been growing significantly.
Overall, the review is well written, covers the topic appropriately and points to relevant gaps in knowledge in a clear manner. The selection of references is appropriate but there are recent references that may be more up to date and would further reflect the state of the field as suggested below.
Minor comments:
1. In section 3.1 the findings of a recent study (Sirenko et al. Blood 2024 Sep 12;144(11):1221-1229) on the prevalence and clinical associations of UBA1 mutations in patients with MDS could be added and discussed.
2. Regarding the chapter on therapeutic approaches in the treatment for VEXAS syndrome (lines 256-266) more recent studies have been published than currently referenced (e.g. Hadjadj et al. Ann Rheum Dis 2024 Sep 30;83(10):1358-1367 and Trikha et al. Haematologica 2024 Oct 1;109(10):3431-3434). The authors should check these newer studies and update the chapter accordingly.
3. Line 275: space between words missing (byUBA1).
4. Line 287: remove the repeat word “patients” in the following sentence “For selected patients younger and fit patients…”.
Author Response
Please find attached our reply.

Reviewer 2 Report
Comments and Suggestions for Authors
Wolff et al. reviews the literature on myelodysplasia and inflammation and how studies of VEXAS syndrome may be useful in understanding development and progression of MDS.
Specific comments:
1. For all table use black to outline the tables. Cannot see the dividing lines for the column headings in the blude shading. Consider increasing the font size for column headings to further denote them as the headings.
2. Reformat table one. The paragraph settings appear to be set to justified which is giving odd spacing of the words in the table.
3. Reference table one more in the text or drop the table. The importance of the table is not coming across in the text.
4. Line 153 Karyotype misspelled.
5. Line 214 It is unclear what reference the “French cohort study” refers to.
6. Reference 45 the author line is garbled.
7. Reference 44 and 23 have google searches as their hyperlink. Provide a direct link to the reference.
Author Response
Please find attached our reply.
